

# The Evolution of Zonally Asymmetric Austral Ozone in a Chemistry Climate Model

Fraser Dennison[1,2,3], Adrian McDonald[1], and Olaf Morgenstern[2,3]

[1]Department of Physics and Astronomy, University of Canterbury, Christchurch, New Zealand
[2]National Institute of Water and Atmospheric Research (NIWA), Lauder, New Zealand
[3]now at NIWA, Wellington, New Zealand

*Correspondence to:* Fraser Dennison (Fraser.Dennison@niwa.co.nz)

**Abstract.**

Asymmetry in the Southern Hemisphere stratospheric ozone hole is important due to both direct radiative heating and its effect on dynamics. It is also a strong indicator of the underlying quality of the stratospheric dynamics of a climate model. We investigate the simulation of the zonal asymmetry in ozone in the NIWA-UKCA atmosphere–ocean–chemistry climate model using elliptical diagnostics, a methodology used for the first time in this subject area. During spring, the region most depleted in ozone is displaced from the pole toward South America based on ERA-Interim and the model output. The model correctly simulates the direction of this displacement but significantly underestimates its magnitude. The model shows that as ozone becomes increasingly depleted over the late 20[th] century this asymmetry in the ozone distribution moves west, before moving east as polar ozone recovers over the course of the 21[st] century. Comparison with model runs in which ozone depleting substances are held fixed at pre–ozone hole levels shows this shift is primarily a function of the magnitude of ozone depletion, although increases in greenhouse gases also have some effect.

## 1 Introduction

In this study, we examine changes in the zonal asymmetry of ozone and the effect this has within a climate model. Until recently, most general circulation models (GCMs) have used zonally averaged ozone distributions (Randall et al., 2007; Hegerl, 2007), but a number of studies have emphasized the importance of modelling the zonal asymmetry of ozone for simulating stratosphere polar temperatures. These studies take the approach of comparing model simulations with zonally averaged ozone to those with either prescribed three dimensionally varying ozone (Crook et al., 2008) or interactively modelled ozone (Sassi et al., 2005; Gillett et al., 2009; Waugh et al., 2009). Crook et al. (2008) found a cooling in the stratosphere and upper troposphere associated with the asymmetric ozone structure. This cooling was located around 150° E, a region with above average ozone, which means that this effect is not due directly to radiative heating, but instead to dynamical heating. Sassi et al. (2005), Gillett et al. (2009) and Waugh et al. (2009) expand on the methods utilized in Crook et al. (2008) by instead using a model with interactively simulated ozone, thus ensuring ozone is consistent with atmospheric dynamics. The findings of these studies are consistent with those of Crook et al. (2008). Additionally, Waugh et al. (2009) notes that the zonally asymmetric cooling effect is larger, when the ozone hole itself is larger, meaning that Antarctic temperature trends are underestimated as a result





of zonally averaged ozone being used in climate models. Sassi et al. (2005) finds that in the mesosphere, where the diurnal cycle in ozone is more prominent, a zonally averaged ozone field will over-estimate day-time ozone and thus the mesospheric temperature.

In addition to the effect on stratospheric temperature, zonally asymmetric ozone distributions also affect tropopause height. Evtushevsky et al. (2008) find that the tropopause height and sharpness are influenced by zonally asymmetric ozone during spring, with the below average ozone regions associated with a higher tropopause and a thicker transition layer. Importantly, thickening of the transition layer between the stratosphere and the troposphere likely increases troposphere–stratosphere exchange and mixing activity.

The distribution of ozone varies with low wavenumber patterns, primarily wavenumber 1 (Ialongo et al., 2012). In the Southern Hemisphere, the asymmetry is typically such that ozone depleted air extends further poleward in the Atlantic region. The asymmetry in ozone during winter and spring is caused by planetary wave driven displacement or distortions of the polar vortex (Waugh, 1997; Ialongo et al., 2012). The polar vortex and the ozone hole are closely linked as ozone depletion affects the temperature gradient and thus the strength of the vortex, while the strength of the vortex affects the degree to which ozone depleted air is isolated from the relatively ozone rich air at mid-latitudes (Beron-Vera et al., 2012; Smith and McDonald, 2014).

The distribution of ozone also feeds back onto the shape of the polar vortex via its effect on the planetary waves. Ozone has been shown to be an important factor modulating the atmospheric refractive index and thus the propagation of planetary waves (Nathan and Cordero, 2007; Gabriel et al., 2007). Gabriel et al. (2007) finds the introduction of zonally asymmetric ozone results in a change in the upward and eastward directed wave flux. Compared to the case of zonally symmetric ozone distributions, these wave fluxes are increased in the western Northern Hemisphere stratosphere and reduced in the east. Albers and Nathan (2012), using a mechanistic model, explore the ways in which zonally asymmetric ozone, via its influence on wave propagation, affects the polar vortex. They identify two pathways in which the polar vortex is influenced. The first describes the effect of zonally asymmetric ozone on vertical energy flux and planetary wave drag, the second describes the effect on zonal-mean temperature and hence thermal wind balance. Both pathways are found to be important in influencing the polar vortex. The incorrect simulation of ozone asymmetries and changes in these asymmetries therefore potentially have widespread impacts on the state of the polar stratosphere.

The ozone asymmetry has exhibited a trend over recent decades. Using observations from the Total Ozone Mapping Spectrometer (TOMS) Grytsai et al. (2005) find that over the period 1979–2003 the position of the ozone minimum at $65°$ S drifts eastward at a rate of $23.6\pm7.2°$ per decade, with the position of the maximum remaining constant. Grytsai et al. (2007) extended this study showing eastward trends in the ozone minimum for all latitudes between 50 and $80°$ S. The cause of this trend (and it's reversal over the subsequent decade) is investigated by Grytsai et al. (2017) by comparing chemistry-climate model simulations in which the concentration of ozone-depleting substances (ODSs) are fixed or to simulations in which varied. They show position of ozone minima to differs latitudinally between the different simulations which suggests that the shifts in the zonal distribution of ozone is due to changes in the concentration of ODSs.

This study examines the changes in the characteristics of the zonally asymmetric ozone. Section 2 contains a description of the model as well as the metric used to describe the ozone asymmetry. Simulation of the ozone asymmetry by the model is



assessed by comparison with reanalysis data in Sect. 3. Section 3 also investigates the evolution of ozone asymmetry in the model from 1950 to 2099 and the relative role of ozone and greenhouse gas (GHG) forcing on zonal asymmetries. Section 4 provides a discussion and summary of the results.

## 2  Data and Methods

### 2.1  Model

The model used in this study is the National Institute of Water and Atmospheric Research–United Kingdom Chemistry and Aerosols (NIWA-UKCA) coupled atmosphere-ocean chemistry-climate model (AOCCM). The model combines an early, low-resolution version of the Hadley Centre Global Environment Model version 3 atmosphere ocean (HadGEM3-AO) model (Hewitt et al., 2011), which includes the Met Office Unified Model (UM) atmosphere model (Davies et al., 2005), the Nucleus for European Modelling of the Ocean (NEMO) ocean model (Madec, 2011), and the Los Alamos sea ice model (CICE) (Hunke et al., 2015), with the NIWA-UKCA chemistry module (Morgenstern et al., 2009, 2017). The UM atmosphere model is non-hydrostatic, fully compressible, and uses a semi-Lagrangian advection scheme (Davies et al., 2005). The version of the UM used here features a number of improvements over the version described by Davies et al. (2005), these are detailed in Hewitt et al. (2011). This model version has a horizontal resolution of $3.75°$ by $2.5°$, a model top at $84\,km$ and 60 model levels. The version of the NEMO ocean model used has a horizontal resolution of $\sim2°$ by $1°$ and 31 model levels.

The NIWA-UKCA model is particularly suited for analysis of the interaction between ozone and climate because it has it's top near the mesopause and a relatively large number of model levels. This is an improvement over previous generations of models, many of which had a model top below the stratopause. The chemistry module includes more than 60 chemical species and around 250 different reactions Morgenstern et al. (2009, 2013, 2017). Importantly, the simulation of ozone is fully interactive, meaning that it responds to atmospheric dynamics as well as chemistry.

In this study we use ensembles of three different model simulations. Our reference simulation has greenhouse gases (GHGs) following the Representative Concentration Pathway 6.0 (RCP6.0) scenario (Meinshausen et al., 2011) and chlorinated or brominated ODSs following the A1 scenario (WMO, 2011). The A1 scenario assumes continued compliance with the Montreal Protocol. This simulation is referred to as "REF-C2" and covers 1950–2100; there are five runs in this ensemble. To isolate the effect of ozone depletion, the REF-C2 ensemble will be compared to a second smaller ensemble of two SEN-C2-fODS simulations; these runs differ from REF-C2 only in that ODSs are held fixed at their 1960 levels, thereby removing the impacts of ozone depletion. Similarly, the effect of GHG forcing is isolated by comparing the REF-C2 runs to a third ensemble consisting of three runs, SEN-C2-fGHG, in which GHGs are held constant at 1960 levels. These simulations are defined by Eyring et al. (2013) and Morgenstern et al. (2017) as part of the Chemistry-Climate Model Initiative (CCMI). These simulations have also been used in other studies such as Morgenstern et al. (2014); Dennison et al. (2015, 2016); Behrens et al. (2016); Oberländer-Hayn et al. (2016).

We also compare this model output to the ozone field from the European Centre for Medium-Range Weather Forecasts (ECMWF) Interim Reanalysis (ERA-Interim) (Dee et al., 2011). ERA-Interim assimilates meteorological and ozone measure-





ments from a variety of sources; the ozone product has been shown to be consistent with independent satellite measurements (Dragani, 2011). The asymmetry in ozone is calculated from total column ozone (TCO) fields. The effect of ozone asymmetry on the stratosphere is shown in the 50 hPa temperature. The 50 hPa level was chosen to approximately match the height of maximum ozone abundance.

## 2.2 Ellipse Fit Metric

The asymmetry of the total column ozone field is examined using metrics based on an ellipse fitted to a monthly-mean TCO contour. Elliptical diagnostics have previously been shown by Waugh (1997) to be useful in examining the polar vortices. In that case, the long–lived tracer $N_2O$ was used to fit the ellipse. Here we use an ellipse fitted to the 300 Dobson Unit (DU) contour of the October mean TCO to describe the distribution of ozone. The 300 DU contour was chosen to ensure the contour would be clearly defined over the entire model simulation (1950–2100). October was chosen because it is representative of the period of peak ozone depletion in both the reanalysis and the model output. The latitude/longitude coordinates of this contour are projected onto a flat plane centred on the South Pole and converted to Cartesian coordinates before fitting an ellipse to the contour using the Taubin (1991) algorithm. The fitted ellipse is defined in terms of the following parameters: angle of rotation, latitude/longitude of the ellipse centre, eccentricity and area. Eccentricity is defined as

$$e = \sqrt{\frac{a^2 - b^2}{a^2}}$$

where $a$ is the length of the major axis and $b$ the minor axis of the ellipse. In the analysis of the centre longitude only data for which the centre is significantly displaced from the pole is used as centres close to the pole have a large uncertainty associated with their longitude. The criterion we will use here is instances where the centre latitude is north of 89° S. Scripts for the calculation of the ozone ellipse is included as a supplement.

## 3 Results

The ability of the model to simulate the asymmetry in ozone can be quantified using the elliptical metric applied to the observational period for comparison with the ERA-Interim ozone output. Figure 1 shows the time series for the October ellipse parameters for the reanalysis and the REF-C2 ensemble. The largest difference between the model and the reanalysis is observed in the latitude of the ellipse centre with the model ellipse centre located much closer to the pole. In the reanalysis, the centre latitude averages to around 80° S although it exhibits considerable variability, ranging between 88° S and 73° S. The model average is approximately 87.5° S, with very little variation. The reanalysis also shows an equatorward trend of 1.2° /decade (which is significant at the 95 % level) during the period 1979–2014; the model shows no such trend over this period. The centre longitude is more consistent between the reanalysis and model; over the period 1979–2014 the mean longitude is slightly further east in the reanalysis, compared to the REF-C2 runs (22° W compared to 37° W), although this difference is not statistically significant. Interestingly, the model shows a change in trend around the mid 1990s which is not evident in the reanalysis. Note however that the reanalysis constitutes only a single ensemble member whereas there are 5 REF-C2 simula-





**Figure 1.** Time series of the October TCO ellipse parameters: (a) centre latitude, (b) centre longitude, (c) eccentricity and (d) area for the REF-C2 (blue) and ERA reanalysis (black). The shaded region indicates $\pm 2$ standard deviations from the ensemble mean.

tions. The turnaround would not be significant in several of the REF-C2 simulations individually. Figure 1(c) shows the ellipse eccentricity. The ellipse is significantly less circular in the reanalysis than the model over this period ($e = 0.48$ compared to $e = 0.38$). There is no trend in either the model or reanalysis eccentricity over this period.

Figure 1(d) illustrates the ellipse area. The modelled ellipse area matches the reanalysis reasonably well. This suggests that the ellipse measure provides a faithful indicator of ozone depletion. Note that the anomalously low TCO during 2002 in the reanalysis is a result of a major stratospheric sudden warming (SSW). This is the only major SSW recorded in the Southern Hemisphere (Baldwin et al., 2003); none are present in the model simulations.

The angle of rotation of the ellipse is not shown here as this parameter exhibits a significant degree of variability in the model due to the very circular nature of the modelled ellipse.



**Table 1.** Trend in the centre longitude ($°$/dec) for each of the model ensembles. Uncertainty is indicated by the 95 % confidence interval

|             | 1960-1999      | 2000-2099     |
| ----------- | -------------- | ------------- |
| REF-C2      | -19.0 $\pm$ 5.1 | 7.6 $\pm$ 1.3 |
| SEN-C2-fODS | -0.8 $\pm$ 8.5  | 2.8 $\pm$ 1.3 |
| SEN-C2-fGHG | -16.2 $\pm$ 7.6 | 6.3 $\pm$ 1.9 |

Figure 2 shows the October ellipse parameters for the REF-C2 , SEN-C2-fODS and SEN-C2-fGHG ensembles over the entire length of the simulation. In order to identify the influence of ozone depletion the Student's $t$ test is applied to the difference between 15 year time slices of the REF-C2 and SEN-C2-fODS ensembles. During periods for which the test shows the difference between the ensemble means is significant ($p < 0.05$) the line is solid, if the test shows no significant difference the line is dashed. Figure 2(a) shows that the centre latitude of the ellipse is similar between all the ensembles, only showing intermittent differences which are likely the result of natural variability. Therefore, indicating no significant influence of either ozone or GHG forcing on this parameter. The central longitude (Figure 2(b)) however, shows a sustained difference between the SEN-C2-fODS and both the REF-C2 and SEN-C2-fGHG ensembles. For the REF-C2 ensemble the period that may be considered significantly different ($p < 0.05$) lasts from around 1980 to the mid 2050s, which corresponds closely to the period in which polar ozone is depleted to levels below 220 DU. Over this period, the difference between the two ensembles is as much as 50°, with the central longitude in the REF-C2 and SEN-C2-fGHG ensembles displaced toward the west.

To further examine the central longitude result, the trends in centre longitude are summarized in Table 1, split into the periods 1960–1999 (representing the period of ozone depletion) and 2000–2099 ( representing the period of ozone recovery). Examination of Table 1 shows that REF-C2 has a trend of $-19°$/decade between 1960 and 1999, which is similar to SEN-C2-fGHG within the bounds of uncertainty. Over this period the SEN-C2-fODS simulation displays no significant trend. This indicates that ozone depletion is the main cause of the shift during this period. Over the 2000–2099 period, SEN-C2-fODS exhibits a small, but significant positive trend. Also, the REF-C2 trend is somewhat larger than that of the SEN-C2-fGHG ensemble. These two results suggest that increasing GHG forcing over the 21[st] century might act in conjunction with ozone recovery to cause the eastward trend in the centre longitude.

For the ellipse eccentricity parameter, Figure 2(c) shows that the SEN-C2-fODS ensemble is relatively consistent at around 0.47; the REF-C2 and SEN-C2-fGHG ensembles also begin around this level. However, during the 1970s, the REF-C2 and SEN-C2-fGHG ellipses become increasingly circular, reaching around $e = 0.4$ and remaining at that level for the remainder of the simulation. It would seem this transition is related to ozone depletion as this change coincides with the beginning of the ozone depletion period. However, it appears eccentricity is only sensitive to ozone up to a certain limit as the REF-C2 and SEN-C2-fGHG ensemble mean trends end well in advance of peak ozone depletion, which suggests saturation of the effect.

Southern Hemisphere ozone has been shown to be related to the Southern Annular Mode (SAM) (Fogt et al., 2009). Here we examine correlations between the ozone ellipse parameters an the 10 hPa SAM (see Dennison et al. (2015) for details on the SAM calculation). We focus on the lower stratospheric SAM here to see the most direct link with ozone if any exists. We



**Figure 2.** Time series of the October TCO ellipse parameters: (a) centre latitude, (b) centre longitude, (c) eccentricity and (d) area for the REF-C2 (blue), SEN-C2-fODS (red) and SEN-C2-fGHG (green). Lines show the ensemble mean smoothed with a 15 year low pass filter, the lines are dashed if the difference between the REF-C2 and SEN-C2-fODS is not statistically significant. The shaded region indicates $\pm 1$ standard deviation from the ensemble mean (the other simulations have a similar level of uncertainty but are not shown for clarity).

find the centre latitude and eccentricity are negatively correlated with the 10hPa SAM index in both the reanalysis ($r = -0.69$ and $r = -0.54$, respectively) and the model ($r = -0.54$ and $r = -0.21$, respectively; all significant at the 95% confidence level). Thus, a positive SAM, and hence a strong vortex, is associated with a more circular ozone hole that is more concentric about the pole. The stratospheric SAM exhibits a positive trend during the historical period associated with ozone depletion

5 (Thompson and Wallace, 2000; McLandress et al., 2010). This could potentially explain the negative trend in eccentricity over this early period. The centre longitude shows no such correlation with the SAM.

Figure 2(d) also shows the time series of the ellipse area. After peaking around 2000, both the REF-C2 and SEN-C2-fGHG ensemble recover close to pre-ozone hole levels by the end of the 21$^{st}$ century. By 2100, REF-C2 has recovered 90% of





the ozone deficit relative to the peak depletion. Eyring et al. (2010) produce similar results using an ensemble of chemistry climate models. It is notable that the recovery of ozone in the SEN-C2-fGHG ensemble is somewhat slower than that of the REF-C2 ensemble. A similar result was also found by Södergren et al. (2016) who show that ozone recovery was hastened by increasingly severe GHG emission scenarios. This effect is due to GHG-induced stratospheric cooling which likely acts to

suppress temperature dependant reactions involved in ozone depletion. Other factors contributing to this speed-up of ozone recovery include an increasing methane abundance, whose net effect is to increase stratospheric ozone, and a speed-up of the Brewer-Dobson Circulation under climate change which is linked to enhanced transport of ozone to high latitudes (Butchart et al., 2006; Garcia and Randel, 2008).

To examine the sensitivity of our results to the analysis methodology, Figure 3 shows the time series of the centre longitude

for ellipses fitted to TCO contours from 220 to 300 DU using the REF-C2 ensemble. This shows that the behaviour shown in Fig. 2 for the 300 DU contour is consistent for other levels, with all contours displaying similar westward trends up until approximately the year 2000 followed by a more gradual eastward trend over the 21$^{st}$ century. However, it is notable that the centre longitude for successively smaller ellipses is located successively further to the east. This result is similar to previous work by Grytsai et al. (2007) who show that the longitude of the ozone minimum is located further east at more poleward

latitudes. This shows that the centre longitude displays a similar tendency if the size of the ellipse is increased either by choosing a larger TCO value for the edge contour or by increased depletion of ozone. It is therefore possible that the centre longitude is only determined by the latitudinal position of the ellipse edge due to a change in the phase of the climatological planetary wave the contour encounters at different latitudes, rather than a change in the ozone modulated atmospheric dynamics. To test this hypothesis, we examine the variation of the centre longitude on the inter-annual timescale to ascertain if it responds

similarly to changes in the size of the ellipse. We find that the centre longitude is, in fact, not correlated with the ellipse area on this shorter timescale. Therefore, we conclude that the cause of these trends in centre longitude are more likely due to effects on atmospheric dynamics such as those described by Albers and Nathan (2012).

We now focus on the effect the ozone asymmetry has on stratospheric climate in the model. In particular, we will examine the centre longitude parameter as this shows the clearest response to both ozone depletion and recovery. Figure 4 shows the

correlation of the centre longitude with the October monthly mean temperature at 50 hPa (the approximate level of the ozone layer) for the reanalysis (top) and REF-C2 runs (bottom) over the period 1979–2014. Significant correlations ($p < .05$) are enclosed by the green contours. For both the reanalysis and model, the correlations show a dipole with positive correlations over the south-eastern Pacific and negative correlations over the Atlantic and Indian Oceans. This shows that an eastward (westward) shift of the ozone hole (which is typically located around 30° W) is associated with positive (negative) temperature anomalies

over the Pacific and negative (positive) temperature anomalies over the Atlantic/Indian Ocean region. This is consistent with a change in radiative heating caused by the shifting of the ozone hole although it could also be the temperature and ozone distribution responding independently to an anomalous phase shift in the planetary wave. This demonstrates that the ozone asymmetry has a clear relationship with temperature and thereby stratospheric dynamics within the model, suggesting the changes in the centre longitude linked to ozone depletion have a wider impact on stratospheric climate.



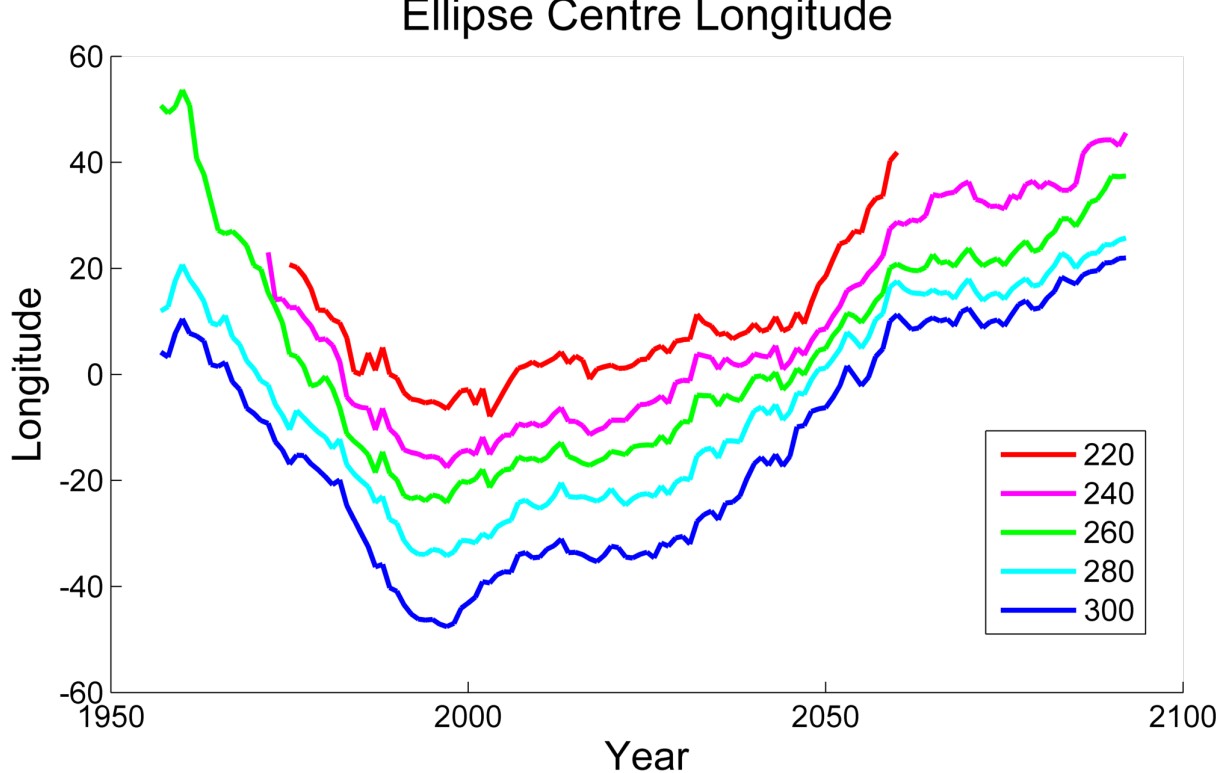

**Figure 3.** Time series of the ellipse centre longitude in the REF-C2 ensemble for different TCO contours

## 4    Discussion and Summary

During spring, the distribution of ozone over polar regions exhibits a significant amount of zonal asymmetry. This asymmetric distribution of ozone is described by an ellipse fitted to the 300 DU contour. This ellipse metric reveals that the model under-estimates this asymmetry in that it simulates a more circular ozone hole with a centre too close to the South Pole. However, the ellipse centre longitude and the ozone hole area are accurately simulated, with both the reanalysis and model showing the ozone hole displaced from the pole toward the Weddell Sea ($\sim$45°W).

Figure 2 shows the centre longitude to be influenced by both ozone depletion and recovery over the 1950-2100 period simulated by the model. During the ozone depletion phase the centre longitude moves west, showing a shift of up to 50° attributable to ozone depletion. As ozone recovery occurs over the 21$^{st}$ century, the centre longitude returns to its pre-ozone hole position. It appears that both ozone recovery and increasing GHG forcing contribute to the eastward trend over this period. This is a somewhat unusual result because ozone recovery (which causes a warming of the polar stratosphere) and GHG forcing (which causes a cooling of the polar stratosphere) generally oppose each other. Opposing influences of ozone recovery and GHG forcing have been shown in other studies with regard to the timing of the vortex breakup (McLandress et al., 2010) and the SAM (McLandress et al., 2011; Morgenstern et al., 2014).




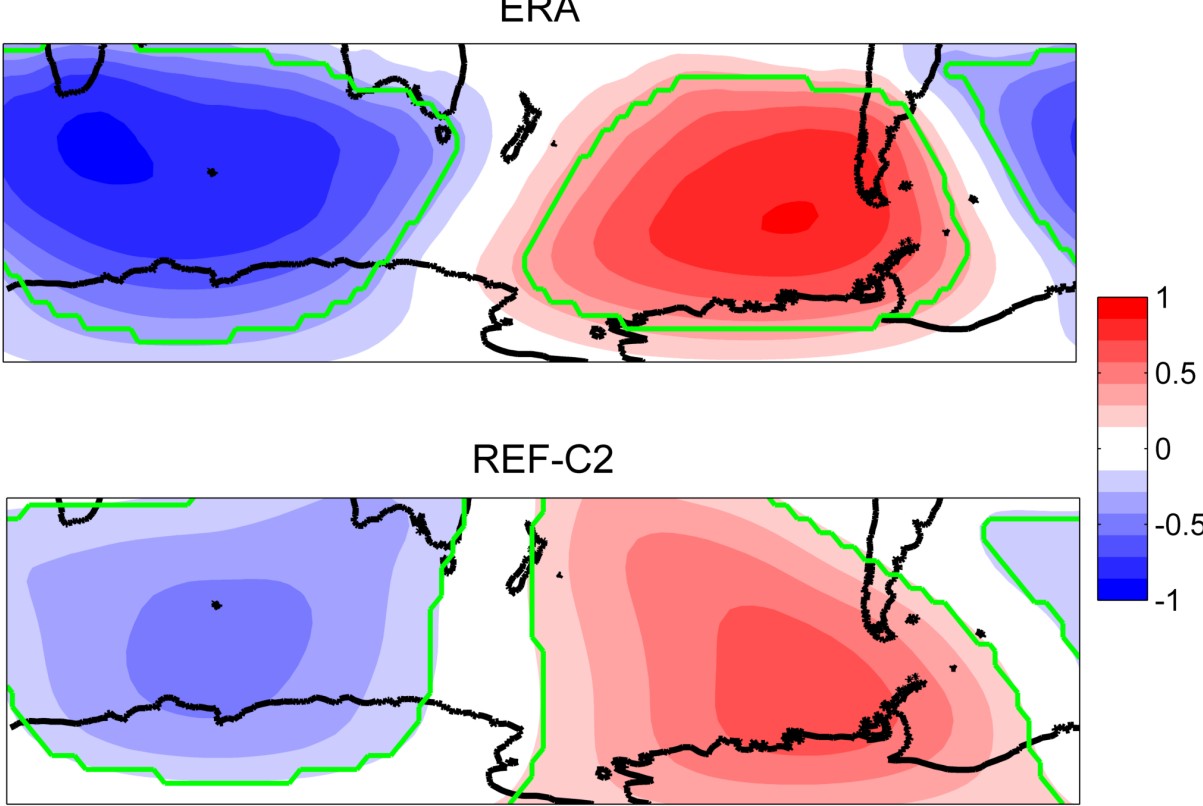

**Figure 4.** Correlation of October central longitude ellipse parameter with the 50 hPa temperature for the ERA-Interim reanalysis (top) and model (bottom)

The ellipse eccentricity is also shown to be influenced by ozone depletion. The behaviour of eccentricity differs slightly from that of the centre longitude in that the influence appears to end around the mid 1980s, 10–15 years prior to peak ozone depletion. This suggests that this effect is saturated as the ellipse approaches the limit of circularity. While $e = 0$ describes a perfect circle, it is possible something near $e = 0.25$ may be close to a practical limit of circularity as this describes an ellipse for which the major axis is only 3 % longer than the minor axis. The eccentricity shows no sign of returning to pre-ozone hole values for much of the 21st century, with the lack of trends in either the SEN-C2-fODS or SEN-C2-fGHG indicating the lack of ozone or GHG forcing. As the effect appeared to become saturated during the 1980s, it is possible ozone may need to recover to 1980 levels before eccentricity is noticeably impacted. In the model, ozone recovers to these levels around 2060; from 2060–2099 the REF-C2 eccentricity does exhibit a small, but statistically significant, trend of 0.02/dec which offers some evidence that this may be the case.

The mechanism driving the changes in eccentricity may be explained via the modulation of the SAM. In both the model and in the ERA-Interim reanalysis, there is an anti-correlation between the 10 hPa SAM and the eccentricity. This is likely due to a positive SAM, and hence a strong polar vortex, inhibiting the upward propagation of planetary waves, thus limiting



the distortion of the vortex and the ozone depleted air it contains. The SAM, which exhibits a positive trend over the ozone depletion period, can thus explain the negative trend in eccentricity. The same cannot be said for the centre longitude, which displays no significant correlation with the stratospheric SAM. Further work is required to investigate the mechanisms driving the trends in this parameter.

The results presented here support those of Grytsai et al. (2017) in that changes in the concentration of ODSs, more so than changes in GHG concentrations, are linked to the shifting zonally asymmetric ozone distribution. However, the approach taken here – the use of elliptical diagnostics – makes the comparison of the particulars of the results somewhat unclear. While this study shows the ellipse centre longitude moves west and the eccentricity decreases over the 1960-1999 time span while Grytsai et al. (2005, 2007) shows the extra-tropical ozone minima to move eastward and the maxima to be stationary. The fact that the

centre longitude moves in the opposite direction to the extra-tropical ozone minima perhaps indicates that it is the rotation of the ellipse that is more descriptive measure of the ozone distribution at mid-latitudes. It is therefore unfortunate that a reliable measure of the rotation was unable to be obtained here due to the models overly-symmetric simulation of ozone.

The shift in the position of the ozone hole also affects climate. In the stratosphere, a shift in the zonally asymmetric ozone is associated with a perturbation in the zonally asymmetric temperature distribution (see Fig. 4). The perturbation is such

that when the ozone hole shifts toward a region, that region experiences a decrease in stratospheric temperature, which could possibly be explained by a decrease in the radiative heating. Previous results (Gabriel et al., 2007; Crook et al., 2008; Gillett et al., 2009; Waugh et al., 2009) have shown that dynamical heating can also be an important factor in the response to zonally asymmetric ozone, but this is not necessary to explain the results in this case. The correlation with the 50 hPa temperature are stronger in the reanalysis compared to the model. Given that the reanalysis features a more pronounced asymmetry in the ozone

distribution it is likely that these measures produce a more distinct relationship with the temperature in this case. The linear regression coefficients are also larger for the reanalysis ($\sim$0.2 K/deg). At 50 hPa, the October temperature is around 210 K, so a $1\sigma$ variation in centre longitude is associated with an anomaly of the order of 3 % in temperature in the reanalysis. In the model, this figure is less than 1 %.

This study uses only a single model, and in particular one for which the magnitude of the zonal asymmetry is significantly

underestimated. It would be of interest to widen the scope of future work to include a ensemble of different models. This could 1) show how common the underestimation of zonal asymmetry is in models 2) possibly confirm the finding that the centre longitude of the ozone ellipse shifts in response to ozone depletion and recovery and 3) show if the sensitivity of the stratospheric climate to these shifts (see Fig. 4) is modulated by the extent to which the model underestimates the zonal asymmetry in ozone.

*Competing interests.*  The authors declare that they have no conflict of interest



## 5  Data Availability

Data from the ERA-Interim reanalysis may be obtained from http://apps.ecmwf.int/datasets/data/interim-full-daily/. The model output has been made available for public use via the Chemistry-Climate Model Initiative (CCMI) archive (see https://blogs.reading.ac.uk/ccmi/badc-data-access/).

5   *Acknowledgements.*  We acknowledge the UK Met Office for use of the Met Office Unified Model (MetUM). This research has been supported by the University of Canterbury; by NIWA as part of its Government-funded, core research; and by the Marsden Fund Council from Government funding, administered by the Royal Society of New Zealand (grant 12-NIW-006). FD acknowledges funding under the Stratospheric Chemistry project of the Deep South National Science Challenge. The authors wish to acknowledge the contribution of NeSI high-performance computing facilities to the results of this research. NZ's national facilities are provided by the NZ eScience Infrastruc-
10  ture and funded jointly by NeSI's collaborator institutions and through the Ministry of Business, Innovation & Employment's Research Infrastructure programme.



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
