# Peer review of "The Evolution of Zonally Asymmetric Austral Ozone in a Chemistry Climate Model"

_Atmospheric Chemistry and Physics, 2017_

## Referee Comment (RC1) · Anonymous Referee #2 · 26 Jun 2017

General comments: In this study the evolution of zonally asymmetric Austral ozone in a specific CCM is examined using elliptical diagnostics for the first time to describe ozone structure in 2 dimensions. The model has some advantages (e.g. direction displacement) and disadvantages (e.g. underestimation of ozone). The model describes the depletion of the 20th century including the westward movement of the asymmetry of ozone and the eastward movement later after ozone starts to recover. These are noval and important findings. It confirms the westward shift of planetary waves induced by zonally asymmetric observed ozone as shown in earlier model studies for the Northern Hemisphere (Gabriel et al. 2007; Peters et al. 2015, 40 years of ERA40 ozone) and for the Southern Hemisphere (Crook et al. 2008). The change of the shift direction after ozone starts to recover is an expected result but the shown evidence that zonally asym-

metric ozone its increase and decrease influences the planetary wave propagation and finally the position of the polar vortex seems for me still very important.

Specific comments: introduction The westward shift of planetary wave due to given ozone waves in earlier studies should be mentioned clearly relative to NH and SH including references including the improved NAM and SAM activity in midwinter.

discussion it should be mentioned clearly that the findings fit very well to former results mentioned above

to Report #2 (iii) authors comment

I do not agree, the shown results for the NH are also relevant for this study, same radiative heating and Dynamics.

recommendation for minor revision after consideration of specific comments

---

## Referee Comment (RC2) · Anonymous Referee #1 · 18 Aug 2017

General comments:

This paper provides a useful advance to the topic through demonstrating the use of elliptical diagnostics to investigate the distribution of ozone in the Southern Hemisphere stratosphere. The authors apply the method to reanalysis data (from ECMWF) and simulations from a particular chemistry-climate model (NIWA-UKCA), fitting an ellipse to the 300 DU contour in total column ozone for October average data to examine the asymmetry of the ozone distribution. They show that ozone depletion has a significant influence on the asymmetry of the ozone distribution over the Antarctic during October, confirming other earlier related work. The main advance is in applying the elliptical diagnostics to the assessment of the Antarctic ozone distribution.

I recommend that the manuscript be published subject to the authors addressing the

following specific comments.

Specific comments:

1. Are formal uncertainties obtained for the fitted parameters? The code in the supplement suggests that formal uncertainties are not calculated, but on reading Taubin (1991), it would seem that this should be possible. It would appear from Figure 3 that the uncertainty on the fitted central longitude is relatively small compared with the interannual variability, but it would be useful to make a statement on the accuracy of the fitting. This could be confirmed by fitting to simulated ellipses of known properties.

2. Related to 1 above, how well-fitted is the 300 DU contour to an ellipse? Not withstanding the reference to Waugh (1997), an ellipse seems rather an arbitrary function to apply to the total ozone column where overburdens and unburdens of ozone due to tilting of the vortex with height will tend to distort the shape of a particular contour. It would be useful to comment on how small and consistent is the difference in area between the 300 DU contour and the fitted ellipse. The authors should also comment on how the tilting of the vortex could influence the fitted parameters. The effect of tilting when the vortex becomes distorted could be checked by fitting to a suitably chosen contours for two partial columns (e.g. 100 hPa and 10 hPa).

3. What is the spatial resolution of the ECMWF data used? How does the spatial resolution of the gridded data influence the accuracy of the ellipse fitting? For example, if the model and reanalysis have different spatial resolutions, how would uncertainties (or spread) in fitted parameters be expected to compare. This could be checked by changing the spatial resolution of the reanalysis data and seeing how the fitted parameters differ compared with those for the original resolution of the reanalysis data.

4. I am unsure how the 2-sigma envelope in Figure 1c for the model ensemble can extend to what looks like 90 degrees south given that the authors state in line 8 of page 4 that they only use fits where the central latitude is north of 89 degrees south. Please clarify. I suggest use of percentiles rather than standard deviations as the

envelope you show is symmetrical about the mean but I would expect the true range to be asymmetric (as, for example, by your method you can't have a central latitude greater than 89 degrees south).

5. The timeseries in Figure 3 run from about 1960 to 2090 which makes me wonder if smoothing has been used (although this is not stated in the caption). Please clarify or indicate why the full time span has not been used.

6. In the discussion, an effort is made to compare with earlier work, and it is noted that the ellipse centre longitude shifts west in the period 1960-1999, while other studies (e.g Grytsai et al., 2007 using data from 1979 to 2005) show that the extratropical zonal ozone minimum shifted eastward in part of this period. However, the authors do not offer an explanation for these differences. Is it possible to reconcile these differences by considering the longitude of the ellipse fitted to the reanalysis data at its most equatorward latitude? From looking at Figure 1 of Grytsai et al. (2007) this would tend to correspond to the longitude of ozone minimum in their analysis, albeit at a somewhat variable latitude that would depend on the size of the vortex (noting that the shift noted by Grytsai et al. (2007) is more eastward towards the pole).

Technical corrections:

Page 2, line 30: its rather than it's

Page 2, line 33: show the position of ozone minima to differ. . .

Page 4, line 9: are rather than is

---

## Author Comment (AC1) · 28 Sep 2017

Thanks for your comments

*Specific comments: introduction The westward shift of planetary wave due to given ozone waves in earlier studies should be mentioned clearly relative to NH and SH including references including the improved NAM and SAM activity in midwinter.*

The following additions (bold) have been made to the text:

(pg 2 ln 8)

Zonally asymmetric ozone has also been shown to have an effect in the troposphere. Evtushevsky et al. (2008) find that the tropopause height and sharpness are influenced

by zonally asymmetric ozone during spring, with the below average ozone regions associated with a higher tropopause and a thicker transition layer. Importantly, thickening of the transition layer between the stratosphere and the troposphere likely increases troposphere–stratosphere exchange and mixing activity. **Crook et al. (2008) find a cooling at the surface over the Ross Sea, although they note there is no significant change to the zonal mean temperature or geopotential height in the troposphere. However, in the Northern Hemisphere, zonally asymmetric ozone has been shown to influence the Northern Annular Mode (NAM) (Peters et al. 2015) and North Atlantic Oscillation (NAO) (Gabriel et al. 2012) with significant impacts demonstrated on surface temperatures, precipitation, wind-driven ocean currents and sea ice thickness.**

(pg 3 ln 3)

**The relationship between ozone and the tendencies of the vortex in the Northern Hemisphere has also been studied by Peters et al. (2015). In this study a CCM simulation forced by ozone the 40-year reanalysis dataset from the European Centre for Medium-range Weather Forecasts (ERA-40) is compared to a simulation forced with zonally symmetrical ozone over the period 1960 to 1999. It was found that the zonally asymmetry, which increases in amplitude over the course of the simulation, leads to a westward shift of the polar vortex.**

*discussion it should be mentioned clearly that the findings fit very well to former results mentioned above*

(pg 11 ln 32)

The results presented here support those of Grytsai et al. (2017) in that changes in the concentration of ODSs, more so than changes in GHG concentrations, are linked to the shifting zonally asymmetric ozone distribution. However, the approach taken here – the use of elliptical diagnostics – makes the comparison of the particulars of the results somewhat unclear. While this study shows the ellipse centre longitude moves

west and the eccentricity decreases over the 1960-1999 time span while Grytsai et al. (2005, 2007) shows the extra-tropical ozone minima to move eastward and the maxima to be stationary. There is some suggestion of an eastward trend shown for ERA-Interim in Figure 1(b) however this is not significant at the 95% confidence level and is substantially smaller than that reported by Grytsai et al. (2005). The fact that the centre longitude moves in the opposite direction to the extra-tropical ozone minima perhaps indicates that it is the rotation of the ellipse that is more descriptive measure of the ozone distribution at mid-latitudes. It is therefore unfortunate that a reliable measure of the rotation was unable to be obtained here due to the models overly-symmetric simulation of ozone. **Interestingly, the results shown here fit with results from the Northern Hemisphere which reveal a westward shift associated with ozone depletion (Peters et al., 2015).**

*to Report 2 (iii) authors comment I do not agree, the shown results for the NH are also relevant for this study, same radiative heating and Dynamics.*

We agree that this is a fair point after reflection. Peters et al. (2015) and Gabriel et al . (2012) have now been added (Gabriel et al. 2007. Is also cited).

---

## Author Comment (AC2) · 28 Sep 2017

Thank you for your comments.

*1. Are formal uncertainties obtained for the fitted parameters? The code in the supplement suggests that formal uncertainties are not calculated, but on reading Taubin (1991), it would seem that this should be possible. It would appear from Figure 3 that the uncertainty on the fitted central longitude is relatively small compared with the interannual variability, but it would be useful to make a statement on the accuracy of the fitting. This could be confirmed by fitting to simulated ellipses of known properties.*

Calculating uncertainty in the parameters shows that, as you suggest, the uncertainty is smaller than the interannual variability. Code for calculating the uncertainty will be

added to the supplement.

Text added (pg 4 ln19):

"The quality of fit of the ellipse was assessed by calculating the fraction of non-overlapping area relative to the overlapping area of the 300 DU contour and ellipse. For both the reanalysis and model the contour is well fitted to the ellipse (median fraction of 0.03 in both cases). The uncertainties were also calculated for the individual ellipse parameters; these were found to be much smaller than the inter-annual variability and so are not shown in the figures included here."

*2. Related to 1 above, how well-fitted is the 300 DU contour to an ellipse? Not withstanding the reference to Waugh (1997), an ellipse seems rather an arbitrary function to apply to the total ozone column where overburdens and unburdens of ozone due to tilting of the vortex with height will tend to distort the shape of a particular contour. It would be useful to comment on how small and consistent is the difference in area between the 300 DU contour and the fitted ellipse. The authors should also comment on how the tilting of the vortex could influence the fitted parameters. The effect of tilting when the vortex becomes distorted could be checked by fitting to a suitably chosen contours for two partial columns (e.g. 100 hPa and 10 hPa).*

The ellipse is well fitted to the 300 DU contour – see above.

Vortex tilt was examined as described in the following addition to the text (pg 5 ln1):

"The Antarctic polar vortex is known to tilt westward and equatorward with height (Waugh1999), hence it is possible using total column ozone may obscure certain characteristics. This was investigated by comparing ellipses fitted to ozone mixing ratio contours at different altitudes (4.5 ppmv at 20 km and 9 ppmv at 26km) chosen to produce ellipses of comparable size to the 300 DU ellipses. It was found that all parameters were strongly correlated between the two levels (r>0.70); hence, the ellipse fitted to the total column should adequately convey the general state of the vortex."
*3. What is the spatial resolution of the ECMWF data used? How does the spatial resolution of the gridded data influence the accuracy of the ellipse fitting? For example, if the model and reanalysis have different spatial resolutions, how would uncertainties (or spread) in fitted parameters be expected to compare. This could be checked by changing the spatial resolution of the reanalysis data and seeing how the fitted parameters differ compared with those for the original resolution of the reanalysis data.*

Added resolution of the reanalysis to the text (pg4 ln9):

Changing the resolution of the reanalysis to that of the model produces no discernible change in fitted ellipse parameters.

*4. I am unsure how the 2-sigma envelope in Figure 1c for the model ensemble can extend to what looks like 90 degrees south given that the authors state in line 8 of page 4 that they only use fits where the central latitude is north of 89 degrees south. Please clarify. I suggest use of percentiles rather than standard deviations as the envelope you show is symmetrical about the mean but I would expect the true range to be asymmetric (as, for example, by your method you can't have a central latitude greater than 89 degrees south).*

It is only the centre longitude corresponding of these fits that are not used. Figure 1(a) does include points South of 89S.

Text has been changed from:

In the analysis of the centre longitude only data for which the centre is significantly displaced from the pole is used as centres close to the pole have a large uncertainty associated with their longitude

To (pg 4 ln 15):

"Ellipses with centres close to the pole have a larger uncertainty associated with their centre longitude. For this reason, only centre longitudes from ellipses with a centre latitude north of 89S are included in the analysis. "

The truncation of the centre latitude distribution at the pole is sufficiently far along the tail of the distribution that the 95th percentile envelope is little different than 2 sigma envelope.

*5. The timeseries in Figure 3 run from about 1960 to 2090 which makes me wonder if smoothing has been used (although this is not stated in the caption). Please clarify or indicate why the full time span has not been used.*

Yes, the time series has been smoothed with a 15-year running mean. This is now stated in the caption to figure 3.

*6. In the discussion, an effort is made to compare with earlier work, and it is noted that the ellipse centre longitude shifts west in the period 1960-1999, while other studies (e.g Grytsai et al., 2007 using data from 1979 to 2005) show that the extratropical zonal ozone minimum shifted eastward in part of this period. However, the authors do not offer an explanation for these differences. Is it possible to reconcile these differences by considering the longitude of the ellipse fitted to the reanalysis data at its most equatorward latitude? From looking at Figure 1 of Grytsai et al. (2007) this would tend to correspond to the longitude of ozone minimum in their analysis, albeit at a somewhat variable latitude that would depend on the size of the vortex (noting that the shift noted by Grytsai et al. (2007) is more eastward towards the pole).*

There is possibly something to this, although the data is not really definitive.

For example, the ERA-Interim ellipses show a 1979-2005 trend in centre longitude of +7 degrees/decade which is not significant at the 95% level (Confidence Interval is [-5,19])

If taking only ellipses with centre latitude north of 83S gives a similar result: +7 degrees/decade (95%CI [-10,23])

Table 2 from Grytsai 2007 shows the trend in the longitude of the ozone minimum at 80 S over this timespan is quite a bit larger at 23.5 +/- 8.8 degrees/dec

Added the bold text in the section below:

(pg 11 ln 32)

The results presented here support those of Grytsai et al. (2017) in that changes in the concentration of ODSs, more so than changes in GHG concentrations, are linked to the shifting zonally asymmetric ozone distribution. However, the approach taken here – the use of elliptical diagnostics – makes the comparison of the particulars of the results somewhat unclear. While this study shows the ellipse centre longitude moves west and the eccentricity decreases over the 1960-1999 time span while Grytsai et al. (2005, 2007) shows the extra-tropical ozone minima to move eastward and the maxima to be stationary. **There is some suggestion of an eastward trend shown for ERA-Interim in Figure 1(b) however this is not significant at the 95% confidence level and is substantially smaller than that reported by Grytsai et al. (2005).** The fact that the centre longitude moves in the opposite direction to the extra-tropical ozone minima perhaps indicates that it is the rotation of the ellipse that is more descriptive measure of the ozone distribution at mid-latitudes. It is therefore unfortunate that a reliable measure of the rotation was unable to be obtained here due to the models overly-symmetric simulation of ozone. **Interestingly, the results shown here fit with results from the Northern Hemisphere which reveal a westward shift associated with ozone depletion (Peters et al., 2015).**

*Technical corrections:*

Page 2, line 30: its rather than it's - fixed

Page 2, line 33: show the position of ozone minima to differ. . . - fixed

Page 4, line 9: are rather than is – fixed